# Food and Food Groups in Inflammatory Bowel Disease (IBD): The Design of the Groningen Anti-Inflammatory Diet (GrAID)

**DOI:** 10.3390/nu13041067

**Published:** 2021-03-25

**Authors:** Marjo J. E. Campmans-Kuijpers, Gerard Dijkstra

**Affiliations:** Department of Gastroenterology and Hepatology, University Medical Center Groningen, University of Groningen, 9713 GZ Groningen, The Netherlands; Gerard.Dijkstra@UMCG.nl

**Keywords:** inflammatory bowel disease (IBD), ulcerative colitis (UC), Crohn’s disease (CD), diet, anti-inflammatory, food, food groups, microbiome

## Abstract

Diet plays a pivotal role in the onset and course of inflammatory bowel disease (IBD). Patients are keen to know what to eat to reduce symptoms and flares, but dietary guidelines are lacking. To advice patients, an overview of the current evidence on food (group) level is needed. This narrative review studies the effects of food (groups) on the onset and course of IBD and if not available the effects in healthy subjects or animal and in vitro IBD models. Based on this evidence the Groningen anti-inflammatory diet (GrAID) was designed and compared on food (group) level to other existing IBD diets. Although on several foods conflicting results were found, this review provides patients a good overview. Based on this evidence, the GrAID consists of lean meat, eggs, fish, plain dairy (such as milk, yoghurt, kefir and hard cheeses), fruit, vegetables, legumes, wheat, coffee, tea and honey. Red meat, other dairy products and sugar should be limited. Canned and processed foods, alcohol and sweetened beverages should be avoided. This comprehensive review focuses on anti-inflammatory properties of foods providing IBD patients with the best evidence on which foods they should eat or avoid to reduce flares. This was used to design the GrAID.

## 1. Introduction

Diet is one of the environmental factors that is associated with the onset and course of inflammatory bowel diseases (IBD) comprising Crohn’s disease (CD) and ulcerative colitis (UC). Accumulating evidence points to a gut dysbiosis combined with an aberrant immune response in genetically predisposed individuals; a process probably triggered and maintained by changes in environmental factors, including diet [1]. The exact interplay between these factors is still unknown. Since the prevalence of IBD is highest in the Western world, affecting up to 0.5% of the general population in 2015 [2], it is thought that the Western diet, high in fats and sugars and low in vegetables and fruits, contributes to the development of IBD [3]. It is established that the Western diet reduces the diversity of the gut microbiome [4].

The strongest proof for the role of nutrition in the aetiology of CD is found in children with IBD, where strict exclusive enteral nutrition (E.E.N.) can induce remission in CD. E.E.N. is currently the primary induction treatment in paediatric CD; partly because steroid therapy has adverse effects and growth has to be maintained in children [5,6]. Since nutrition is expected to play a role in the pathophysiology of IBD, effectiveness of several other (partial) whole food diets in IBD were studied in order to know which foods to include or avoid. Chiba et al. demonstrated that a semi-vegetarian diet was more effective in maintaining remission in CD patients than an omnivorous diet [7]. The IBD-AID, an anti-inflammatory diet that restricts intake of certain carbohydrates and includes pre- and probiotic foods and modifies dietary fatty acids [8] showed some improvements in a case series study as adjunct dietary therapy for treatment of IBD patients. Besides, the specific carbohydrate diets (SCD), initially developed to treat children with celiac disease, thus eliminating all grains, was trying to shift the microbiome in IBD patients [9]. A recent RCT in children with mild/moderate CD revealed that SCD had similar remission rates compared to a whole food diet [10]. Recently, partial enteral nutrition combined with a Crohn’s Disease Exclusion Diet (CDED), demonstrated to be a useful strategy in both children and adults in whom biological therapy failed [11]. Additionally, a paediatric RCT demonstrated that partial enteral nutrition combined with CDED was equally effective but better tolerated than E.E.N in mild to moderate CD [12]. The Crohn’s Disease TReatment with EATing (CD-TREAT) (CD-TREAT) diet, a food-based diet with similar composition to E.E.N., replicates the effectiveness of E.E.N. and was better tolerated in children with active CD [13]. Sometimes, a diet low in fermentable oligosaccharides, disaccharides, monosaccharides and polyols (FODMAP) diet is also used in IBD to reduce gastrointestinal (GI) symptoms, but it failed to demonstrate effectiveness in patients with quiescent IBD and even reduced their faecal abundance of gut microbes [14,15].

As patients with IBD often experience complaints of certain foods and are eager to postpone new flares, they frequently ask “What should I eat?” [16]. However, current guidelines are lacking, since the majority of dietary studies are prone to recall bias [17] or other methodological issues [17,18,19] and results on dietary effects are too inconsistent to be applicable in clinical practice [20,21].

Thus, IBD patients, who are often already malnourished [22,23], start experimenting with their food to alleviate symptoms. However, when these diet modifications are not properly supported by a dietitian or physician, this can lead to overly restrictive diets and may even further compromise the patient’s nutritional status and negatively affect their disease outcomes [24,25,26]. Therefore, both patients and physicians are in desperate need of evidence for an anti-inflammatory dietary advice in CD. 

Studying diet is complex as diet consists of multiple foods and nutrients that will be consumed in varying combinations and quantities, which can also interact with each other. Therefore, it will take many decades until every single component and food item will be proven beneficial or detrimental. Furthermore, diet can be studied on several levels: dietary patterns, food level and nutrient level. To develop an effective anti-inflammatory dietary intervention for IBD patients, we first need to understand the role of diet in aetiology and pathology in IBD at all three levels. Here a review is presented on food (group) level.

## 2. Materials and Methods

In this narrative review, literature was studied regarding the effects of food and food groups on the onset and course of IBD. Dietary research is complex and is often flawed [17,18,19]. No systematic review was performed since too many different food items or food groups are involved in this topic. Besides, we are aware that not on every single food or food groups sufficient evidence can be found, but by reviewing literature and gathering this information in the Groningen anti-inflammatory diet (GrAID), we want to provide the patients with the best evidence there is. In this review the evidence on aetiology and pathology in IBD patients was key, followed by evidence in other human trials, evidence on changes in the microbiome and IBD animal models or in vitro models. 

Information on GI complaints was not the subject in this review because these complaints can vary between individuals and patients might decide for themselves, based on specific situations or experiences, to avoid these products or not. 

Based on this evidence the GrAID was designed and compared to other existing diets in IBD on food (group) level.

## 3. Results

### 3.1. Results on Foods and Food Groups

#### 3.1.1. Meat

Meat is an excellent source of protein, iron and vitamin B12. Furthermore, meat contains saturated fat. Meta-analyses of prospective epidemiologic studies showed that there is no significant evidence for concluding that dietary saturated fat is associated with an increased risk of cardiovascular disease [27,28]. However, recent analyses of two prospective cohort studies in healthy US people showed that an increase in total red meat consumption of at least half a serving per day, especially processed meat, was associated with higher mortality rates [29]. A prospective study including 412 patients with quiescent UC treated with an aminosalicylate, reported that dietary intake of myristic acid, a saturated fatty acid was associated with increased risk of a flare (Odds Ratio (OR) 3.01; 95% CI: 1.17–7.74) [30]. Meat and processed meat also contain high levels of organic sulphur and sulphate additives, which may increase the amount of sulphate for microbial produced hydrogen sulphide. End-products of protein fermentation, particularly H_2_S, ammonia and to a lesser extent, phenols have well-established detrimental effects on the colonic microenvironment and epithelial health [31].

In French middle-aged women, high total protein intake, specifically animal protein, was associated with a significantly increased risk of IBD, (hazards ratio for the third vs. first tertile being 3.31 (95% CI: 1.41–7.77; *p*_trend_ = 0.007) for total protein, and 3.03 (95% CI: 1.45–6.34; *p*_trend_ = 0.005) for animal protein [32]. Among sources of animal protein, high consumption of meat or fish but not of eggs or dairy products was associated with IBD risk [32]. 

Meat intake was positively associated with disease relapse in IBD. The highest quartile for meat consumption coincided with a higher risk of active disease (OR 3.61; 95% CI: 1.15–11.38). Though, after adjusting for anti-TNF medication, the significance was lost [33]. Red meat is defined as all meat from livestock (lamb, mutton, beef, pork, veal, goat, horse). Processed meats are red or white meat that was prepared with smoking, salting (adding salt enriched with nitrates and nitrites), curing or addition of preservatives [34].

An RCT in adults with CD in remission showed that in the high meat group (consumption of two or more servings/week of red or processed meat for 20 weeks) a moderate to severe relapse occurred in 62% of the patients with CD, whereas in the low-meat group (with not more than 1 serving per month) the relapse rate was only 42.4%. This difference was not significant [35]. Limitations were the lower adherence to the low meat diet and not being clear what was used as replacement. Furthermore, only one-third of patients (more in high meat group) who were randomised signed the consent form. Besides, the difference in meat consumption might be needed to be more extreme.

Recently a prospective cohort study reported that a carnivorous dietary pattern comprising poultry, processed meat and red meat was associated with a higher risk of UC development (OR 1.11; 95% CI: 1.01–1.22) [36]. Another prospective study evaluating dietary intake and risk of relapse of UC, showed that red and processed meat had the strongest association with risk of relapse of UC (OR 5.19; 95% CI: 2.09–12.9) [37].

In the CDED diet, an exclusion diet given to patients with CD who have flares, red meat is not allowed, lean meat is allowed and at least 150–200 g chicken daily is mandatory [12,38].

The faecal microbiota of vegetarians and vegans showed lower abundance of *Bifidobacteria* and Bacteroides species than omnivores living on the same continent [39]. In a 2-year prospective trial, IBD patients following a semi-vegetarian diet 15/16 maintained remission compared with 2/6 on a regular diet (*p* = 0.0003) [7]. This semi-vegetarian diet is lacto-ovo-vegetarian with fish once a week and meat once every two weeks.

#### 3.1.2. Fish

Fish oil contains long chain n-3 polyunsaturated fatty acids (PUFAs), eicosapentaenoic acid (EPA) and docosahexaenoic acid (DHA) which have demonstrated anti-inflammatory properties in several chronic inflammatory disorders such as IBD. They are involved in the regulation of immunological and inflammatory responses [40]. Despite clinical trials of fish-oil derivates showing beneficial effects in IBD implying biological plausibility, a systematic review found conflicting data on the association of fish with CD risk (OR range 0.46–2.41) [41]. Three of the four studies included in this review reported that higher fish and seafood consumption was associated with a higher UC risk. However, these studies failed to demonstrate an association with reduced clinical relapse. Therefore, there is insufficient evidence to recommend the use of n-3 PUFAs in clinical practice [42]. No statistically significant associations were found between the intake of fish and disease relapse in IBD [33]. Among the sources of animal protein, high consumption of meat or fish/sea products was even associated with IBD risk (*p*_trend_ = 0.05) [32]. In DNBS-induced colitis in rats, administration of canola oil, safflower oil and fish oil were compared. Fish oil showed the highest body weight and the least histological damage [43]. In the IBD-AID fish can be used [8].

#### 3.1.3. Tofu and Soy Protein

Tofu was negatively associated with body mass index (BMI), body fat mass and body trunk mass in IBD patients (R = −0.019; *p* < 0.05) [44]. In DSS-induced colitis in pork, a soy derived hydrolysate enriched in di- and tripeptides, exerted anti-inflammatory effects, prevented gut permeability and expression of inflammatory cytokines in the colon [45]. In DSS-induced colitis in mice, treatment with VPY, a novel PepT1-transported tripeptide derived from enzymatic hydrolysis of soy glycinin, down-regulated the expression of pro-inflammatory cytokines, reduced inflammation severity and DSS-induced colitis symptoms [46]. VPY from soy products may be a potential therapeutic agent in IBD. A recent study in IBD mouse models demonstrated that replacement of animal protein in a Western diet by plant-based sources such as soy and pea protein reduced IBD severity in all IBD mouse models [47].

#### 3.1.4. Eggs

A prospective study in French middle-aged women showed that high animal protein intake but not eggs or dairy products, was associated with the incidence of IBD risk [32]. There was no clear *p* for trend in this study (*p*_trend_ = 0.91). A significantly enhanced immune response to egg white/egg yolk was observed in CD patients [48]. 

In a pilot study in 40 CD patients IG4 antibodies to 14 specific food antigens such as egg white, egg yolk, potato, tomato, cheddar cheese, rice, beef, lamb, pork, soya, peanuts, wheat, yeast and chicken were tested, and each subject excluded the four most reactive foods for 4 weeks. In the 29 patients who completed the study, a significant reduction on a modified Crohn’s Disease Activity Index and a reduction in the ESR as compared to pre-treatment levels was found. However, a control group was lacking [49].

A systematic review found conflicting data on the association of eggs with CD risk (OR range 0.4–1.76) and four out of five studies found a higher UC risk [41]. 

In the CDED diet, an exclusion diet for CD-patients with active disease which was associated with high rates of remission, consumption of 2 eggs a day is required [12,38].

In DSS-induced colitis in mice, digestion of preserved egg white ameliorated clinical symptoms, such as weight loss, disease activity index and inhibited secretion of pro-inflammatory cytokines TNF-α and IL-6 [50].

#### 3.1.5. Dairy

Dairy foods are an important source of protein, riboflavin, and the main dietary source of calcium. Calcium is necessary to prevent metabolic bone disease in IBD patients [51]. However, milk products also contain lactose. If lactose is not absorbed, due to lactase deficiency, this reaches the colon, where it may cause bloating and/or diarrhoea. However, the prevalence of lactase-deficiency is not higher in IBD patients than in healthy controls [51]. Thus, unless it clearly worsens symptoms, IBD patients should not limit their milk consumption during flares. Furthermore, they may still tolerate dairy foods with less lactose (such as yogurt) or without any lactose (such as hard cheeses). 

Consumption of dairy product was neither associated with IBD risk (*p*_trend_ = 0.93) [32] nor with disease relapse in IBD [33]. A systematic review on dairy and UC development reported an OR between 0.79 and 2.67 [41]. 

Although, fermented dairy products are associated with reduced overall mortality, reduced risk of impaired glucose metabolism and positive effects on blood pressure and total cholesterol, the effects of consuming fermented foods in IBD have yet to be established [52]. Animal studies suggest favourable effects of fermentable foods in intestinal barrier integrity. 

In the CDED, dairy products are prohibited [38], in the IBD-AID and Low FODMAP diet they are allowed in limited amounts [8,53]. The SCD eliminates all milk products except for hard cheeses and fermented yogurt [9].

##### Milk

Looking at symptoms in CD patients, sheep and goat’s milk were less likely to enhance symptoms than cow’s milk [54]. Soy milk, although not strictly a dairy product, consistently appeared better tolerated than milk products. Yogurt was also better tolerated than milk, although they did not differentiate between different fat contents. This positive response to yogurt may possibly reflect a response to probiotics. In a small dietary intervention trial in 16 patients with CD in clinical remission, cow’s milk, apple and wheat products were reported as the most frequent dietary symptom triggers [55].

##### Yoghurt

Lorea Baroja et al. demonstrated an anti-inflammatory effect of probiotic yogurt in 20 healthy patients and 20 subjects with IBD (15 with CD and 5 with UC). After 1 month, the proportion of CD4^+^CD25^high^ T-cells increased in the IBD patients (*p* = 0.007) [56].

Consumption of experimental fermented dairy product like yogurt and cheese demonstrated anti-inflammatory effects in mice colitis model and might have potential anti-inflammatory effects in IBD patients [57]. Also, soy milk fermented with Lactococcus lactis S-SU2 demonstrated anti-inflammatory properties in DSS-induced IBD model mice [58].

##### Kefir

Milk kefir is a fermented dairy product, in many ways similar to yogurt. 25 IBD patients (10 CD and 15 UC) consumed 400 mL of kefir daily for 4 weeks. Compared to 20 control IBD patients, in CD patients, sedimentation rate and C-reactive protein decreased significantly, and haemoglobin increased. Moreover, during the last 2 weeks, bloating scores decreased significantly (*p* = 0.012), and feeling good scores enhanced (*p* = 0.032) [59]. However, in UC patients, consumption of kefir only led to significant increase in Lactobacillus (*p* = 0.001). In a double blind RCT, the additional effect of 500 mL kefir vs. 250 mL daily milk for 2 weeks was tested in 82 H pylori patients treated with antibiotics. It showed that in patients treated with kefir H pylori was eradicated in 78% (vs. 50%; *p* = 0.026) and these patients had a lower occurrence of diarrhoea, abdominal pain and nausea [60].

##### Cheese

Emmenthaler Cheese

An animal study in DSS-induced colitis in mice revealed that consumption of Emmentaler cheese, fermented by *Propionibacterium freudenreichii* and another specially developed cheese with a single strain of this bacteria, enhanced DSS-induced colitis and histological scores and reduced weight loss and disease activity index. Both cheeses reduced IgA secretion in the small bowel and enhanced occluding gene expression preventing induction of TNFα, IGNγ and IL-17 [61].

Mozzarella Cheese

Buffalo milk contains many bioactive peptides that are mainly released during digestion of milk (products). A peptide from buffalo milk-derived products (MBCP) found in Buffalo Mozzarella Cheese was able to reduce H_2_O_2_-induced oxidative stress in intestinal epithelial cells and erythrocytes. Both in vitro (on inflamed human intestinal Caco2 cells) and in vivo (in dinitrobenzene sulfonic acid (DNBS) mice model of colitis), non-toxic concentrations of MBCP enhanced adherent epithelial junctions’ organisation, regulated the NF-kB pathway and reduced intestinal permeability. Oral MBCP administration decreased intestinal inflammation in the DNBS-induced colitis. Therefore, consumption of buffalo mozzarella cheese might be beneficial [62]. 

#### 3.1.6. Fruit

Apart from sugars, fresh fruits are generally high in fibre and anti-oxidant vitamin C. Fruits also contains phenolic acids which are absorbed through intestinal tract wall and have potential anti-oxidative and anti-inflammatory properties [63]. The highest concentrations chlorogenic acid, a phenolic acid, are found in apples, blueberries, citrus fruits, mangos, kiwis and plums [64]. Raspberry seed flour, also containing phenolic acid, attenuates high-sucrose diet-mediated hepatic stress and adipose tissue inflammation [65].

Daily consumption of fruit decreased the odds for CD (OR 0.39; 95% CI: 0.22–0.70) and UC (OR 0.56; 95% CI: 0.33–0.95) [19]. An earlier systematic review on fruit and CD development also including case control studies, included five studies that all reported a significant association between fruit intake and CD development [41]. The first included study comparing children consuming fruit over four times a day to those consuming fruit less than once a day found an OR of 0.58 (95% CI: 0.37–0.91) [66]. Furthermore, twins with CD had lower fruit intakes compared to their co-twins (OR 0.2; 95%CI: 0.1–0.9) [67]. Finally, a case-control study demonstrated an OR of 0.49 (95% CI: 0.25–0.96) comparing the highest and lowest quartile of fruit intake and CD development in children [68]. 

A more recent meta-analysis on four prospective studies [23,67,69,70] demonstrated a significant inverse association between fruit consumption with CD risk (Relative Risk (RR) 0.47; 95% CI: 0.38–0.58; I^2^ = 32%) [71]. This association remained significant when analysed in subgroups in servings per day. 

The earlier systematic review on fruit and UC development that also included case-control studies, reported that none of eight included studies in the systematic review showed significant results (OR range 0.42–2.9) [41]. In the later systematic review including the same four earlier mentioned studies, a pooled analysis demonstrated that fruit intake (g/day) was significantly associated with UC risk (RR 0.69; 95% CI: 0.55–0.86; I^2^ = 87%) [71]. However, this association lost significance when it was performed in 1–3 servings/day in subgroups. 

Although citrus fruit consumption was negatively associated with the development of CD (OR: 0.5; 95% CI: 0.3–0.7) [72], grapefruit consumption commonly leads to an enhancement of symptoms, causing significant distress to many IBD patients [54]. 

Besides in a pilot study in IBD patients, 200–400 g of mango pulp (Magnifera indica L) for 8 weeks demonstrated improved Simple Clinical Colitis Activity Index (SCCAI) score and lower levels of pro-inflammatory cytokines including IL-8 [73]. Mango, rich in gallotannin, also enhanced the faecal microbial composition in those patients. In DSS-induced colitis in mice, mango attenuated symptoms through inhibition of NF-κB and MAPK signalling [74].

An RCT in healthy individuals with infrequent stool habits and low fibre intake, showed that prunes significantly increased *Bifidobacteria* across the groups (*p* = 0.046) [75]. There were no significant differences in any of the bacteria measured. Besides, prunes increased stool weight and frequency and were well tolerated but had no effect on SCFA or stool pH.

Bananas contain inulin, one of the best-known prebiotics. Green bananas contain large amounts of type 2 granular resistant starch [76]. Resistant starch is a high amylose starch which can acts as a prebiotic ingredient [77]. This non-digested resistant starch may be fermented into SCFA by colonic bacteria [78]. 

One apple and two bananas a day are mandatory in the CDED diet, an exclusion diet for CD-patients with flares [12]. Furthermore, avocado, strawberries, melon, pear, peach, kiwi and blueberries are allowed. After 10 weeks all fruits are allowed, except dried fruits.

#### 3.1.7. Vegetables

Vegetables contain dietary fibres and are an important source of antioxidant vitamins A, C and E, minerals, trace elements and phenolic compounds. The anti-inflammatory properties of these phenolic compounds can be used as a natural source for the prevention of IBD [79]. An earlier systematic review found conflicting results on vegetable intake and CD risk [41]. They included a case-control study, in which an OR of 0.69 (95% CI: 0.33–1.44) was found between the highest and lowest quartile of vegetable intake and CD development in children [68]. Patients consuming more vegetables were less likely to have CD (OR 0.85 95% CI: 0.74–0.97) [44]. A more recent meta-analyses including three prospective studies [23,69,70] demonstrated a significant inverse association of vegetable consumption (g/day) with odds of CD (RR 0.52; 95% CI: 0.46–0.59; I^2^ = 79%) [71]. 

A systematic review on vegetable intake and UC risk also including case-control studies included eight studies [41]. Three studies reported that a high vegetable intake was associated with decreased UC risk but statistical significance was lacking [80,81,82]. A recent meta-analyses on only prospective studies, included three prospective studies [23,69,70] demonstrating a significant inverse association of vegetable consumption (g/day) with odds of UC (RR 0.56; 95% CI: 0.48–0.66; I^2^ = 72%). 

No statistically significant associations were found between disease relapse and the intake of vegetables in IBD patients [33].

In artichoke, asparagus and onions, oligofructose and inulin, polymers of fructose are found [83] that increase commensal faecal and mucosal *bifidobacteria* and *F prausnitzii* in healthy humans [84].

##### Cruciferous Vegetables

It is known that cruciferous vegetables, such as kale can cause bloating in people with IBS. However, cruciferous vegetables such as broccoli, Brussels sprouts, cauliflower, cabbage and kale do contain indole-3-carbinol which activates the aryl hydrocarbon receptor (AHR). In mice models, consumption of broccoli increased intestinal AHR activity, reduced microbial abundance of *Erysipelotrichaceae* and impaired colitis [85]. 3,3 Diindolylmethane (DIM), also originating from cruciferous vegetables significantly increased the transepithelial electrical resistance of human cultured Caco-2 cells, thereby restoring the intestinal permeability [86]. 

##### Tomato

Tomato is a frequently seen food intolerance in patients with IBS [87]. A significantly enhanced immune response to tomato was observed in CD patients [48]. However, in the CDED diet, an exclusion diet for CD-patients with flares associated with high rates of remission, consumption of two tomatoes and two peeled cucumbers a day are mandatory during the first phase; and tomatoes, cucumbers, carrots, fresh spinach, lettuce leaves, Zucchini are allowed in the first 10 weeks [12]. After 10 weeks all vegetables are allowed, except frozen vegetables, kale, leeks, asparagus and artichoke. 

##### Mushrooms

Since ancient times, mushrooms have been applied as medicinal therapy to prevent inflammation. In IBD patients, a mixed extract of basidiomycetes mushrooms alleviated the inflammatory symptoms [88]. Mushroom glucan exerts a beneficial effect on IBD as it modulates cytokine profiles and phagocyte activity, enhances protection against inflammation, infections and sepsis [89].

##### Corn

Although tolerated by very few, corn eaten as a vegetable was reported by 45% of the CD patients causing adverse effects. Negative results were also associated with other forms of corn, i.e., corn flakes, corn crackers and popcorn [54]. These data are consistent with previous reports [90] and identify corn as one of the key dietary items that may need to be avoided by some individuals with CD. This could be related to fructan composition, or to mechanical properties of the testa, the outer covering of the seed. The distinctive polyphenol component of corn might also relate to the adverse properties associated with corn in all these studies [91]. 

##### Sweet Potato

Worldwide, sweet potato (*Ipomoea batatas* L.) is regarded as the seventh most important food crop [92]. Purple sweet potato is a member of the sweet potato family and is a highly nutritious vegetable; its tubers contain vitamins such as tocopherol and β-carotene, amino acids, minerals and dietary fibre [93]. It also contains phenolic acids, anthocyanins and polyphenols, which have anti-inflammatory activity [94]. Its polysaccharide also has anti-inflammatory properties [95,96]. 

##### Legumes

Patients consuming more serving of beans/legumes were less likely to have UC (OR 0.66; 95% CI: 0.46–0.96) [44]. A cross-sectional study in 103 adult IBD patients (50 with active disease and 53 in remission) showed protective effects of potatoes aggregated with legumes. In this study legume and potato consumption was inversely associated with disease relapse (*p*_trend_ = 0.023) with patients in the highest quartile for legume, and potato consumption carrying a 79% lower risk of active disease (adjusted OR 0.21; 95% CI: 0.57–0.81) compared to those in the lowest quartile [33]. Legumes might potentially aggravate symptoms, but are allowed in the second phase of the CDED diet, an exclusion diet for CD patients with flares [12,38]. The SCD legumes (i.e., lentils, split pea) are permitted, however others (i.e., chickpeas, soybeans) are not [97]. Consumption of red kidney beans, rich in dietary fibre and resistant starch, demonstrated a substantial effect on gut microbiota and caecal fermentation in rats [98].

##### Potatoes

Skins of potatoes contain glycoalkaloids (solanine and chaconine), which can permeabilise cholesterol-containing membranes, and lead to disruption of epithelial barrier integrity. Frying potatoes concentrates glycoalkaloids. Whereas two studies in mice demonstrated that glycoalkaloids increased intestinal permeability in a dose-depending way [99,100], a cross-sectional study in 103 adult IBD patients (50 with active disease and 53 in remission) showed protective effects of potatoes aggregated with legumes. In this study legume and potato consumption was inversely associated with disease relapse (*p*_trend_ = 0.023) with IBD patients in the highest quartile for legume, and potato consumption carrying a 79% lower risk of active disease (adjusted OR 0.21; 95% CI: 0.57–0.81) [33]. However, in this study potatoes were aggregated with legumes. Furthermore, potatoes contain high levels of resistant starch. The prebiotic effect of this highly resistant starch is further increased by cooking and subsequent cooling [101].

In the CDED diet, daily consumption of two fresh potatoes, peeled cooked and cooled before consumption is mandatory [12,38]. In the SCD diet potatoes and yams are not allowed [97].

##### Cereals

No statistically significant associations were found between the intake of cereals and disease relapse in IBD [33]. Patients with CD were less likely to consume whole grains (OR 0.08; 95% CI: 0.70–0.93) and both patients with CD (OR 1.27; 95% CI: 1.11–1.45) and patients with UC (OR 1.26; 95%CI: 1.10–1.44) were also more likely to consume more refines grains [44]. A case control study of 267 incident IBD cases reported that daily muesli consumption decreased the odds for CD (OR 0.15; 95% CI: 0.05–0.51), but not significant in UC (OR 0.50; 95% CI: 0.24–1.03) [19].

##### Maize

A significantly enhanced immune response to maize was observed in CD patients [48]. 

##### Wheat

A case-control study of 267 incident IBD cases and 267 matched controls showed that daily intake of whole grain bread significantly reduced the odds for CD (OR 0.26; 95%CI: 0.15–0.48) and UC (OR 0.42; 95% CI: 0.26–0.70) [19]. Wheat bran contains 46% of non-starch polysaccharides (NSP) from arabinoxylans, cellulose and β-glucans [102]. Wheat bran was well tolerated and could be beneficial for IBD patients, but not replace medicinal treatment [103]. Furthermore, it protects against colon cancer, a type of cancer, which often appears in IBD patients [104,105]. A randomised crossover trial comparing wheat bran to psyllium in children with quiescent UC treated with sulfasalazine demonstrated that only wheat bran supplementation reduced bile acid measurements by 55% (*p* < 0.05) [106]. These bile acids are needed for digestion and absorption of fats and fat-soluble vitamins in the small bowel [107].

However, wheat might also trigger GI symptoms. In a small exploratory study in 16 patients with CD in remission, dairy and wheat products were reported as the most frequent dietary product to trigger GI symptoms [55]. Wheat also contains gluten (gliadin) which may aggravate GI symptoms such as bloating, abdominal pain and diarrhoea, independent of celiac disease [108,109]. Although gluten sensitivity arises with celiac disease, IBD patients also commonly report gluten sensitivity; 28% IBD patients in a British cohort study [110] and 24% of CD and 27% of UC patients in an American study [111]. This self-reported gluten sensitivity was strongly associated with having had a recent flare (OR 7.45; 95% CI: 1.63–34.14; *p* = 0.01). This suggests that gluten sensitivity might be a transient phenomenon in some patients [111]. In an American study, 8% of IBD patients actively used a gluten-free diet to control symptoms, which was associated with improvement of fatigue (*p* < 0.03) [112]. In healthy volunteers, a four-week gluten-free diet decreased *Veillonellaceae* [113]. Higher abundance of *Veillonellaceae*, which is considered to be a pro-inflammatory family of bacteria, is consistently reported in CD patients [114,115]. This increase leads to changes in the pathways for butyrate, tryptophan and fatty acid metabolism [113]. 

In a mice model of ileitis, gluten aggravated inflammation independent of its effects on the microbiota [116]. In non-obese diabetic mice fed with gluten compared with standard chow, gluten reduced intestinal Tregs [117]. In ex vivo human duodenal biopsies, gluten increased the intestinal permeability [118].

Besides, wheat also contains amylase trypsin inhibitors promoting intestinal inflammation by TLR4 activation on myeloid cells [119,120]. Furthermore, amylase trypsin inhibitors have a detrimental effect on the intestinal microbiota promoting intestinal inflammation in mice [121].

In the IBD-AID, SCD diet, CDED diet, CD-treat diet, wheat and gluten consumption is not allowed [8,12,13,38,122].

##### Oat

Oat is a cereal containing carbohydrates including dietary fibre and beta-glucans. In contrast to wheat, barley and rye, oats do not contain gluten. In 1978, a study in 39 UC patients administering 25 g oat per day concluded that high bran intake is of less value in maintaining clinical remission in patients with UC compared to a drug group [103]. However, they overlooked the difference in UC patients; oat bran may be useless for patients with severe symptoms, but it may be effective for patients with mild symptoms, and the dose of oat bran was insufficient compared to other studies [89].

In 22 patients with quiescent UC, 3 months of daily administration with 60 g oat bran (main composition is 1,3- and 1,4-glucan) showed an increase in faecal butyrate by about 36% [123]. Besides, abdominal pain and gastrointestinal reflux improved significantly after the oat bran intervention. Because the intervention did not increase relapse of the disease or GI complaints, patients with quiescent UC can safely use a diet rich in oat bran specifically to increase their faecal butyrate levels. 

In lipopolysaccharide (LPS)-induced enteritis rats, supplementation with oats-glucan with high and low molecular weight revealed that supplementation with both glucan fractions significantly reduced blood leucocytes, but only oats glucan with high molecular weight reduced the lipid peroxidation [124].

In dextran sulphate (DSS)-induced colitis mice, oat β-glucans significantly reduced clinical symptoms of mice, like weight loss, diarrhoea and increased the colon length [125]. The reduced disease activity index and degree of histological colon damage showed that the severity of colitis improved by oat β-glucans. The expressions of pro-inflammatory factors such as TNF-α, IL-1β and IL-6 in the colonic tissues were downregulated.

##### Rice 

Analysis of Japanese epidemiological data suggested that the registered number of patients with CD or UC started to increase more than 20 years after a decreased consumption of rice [126]. A significantly enhanced immune response to rice was observed in CD patients [48].

Unrefined rice containing food fibres is known to increase the beneficial bacteria and reduce the number of Clostridium species [127]. 

In the CDED diet consumption of rice is mandatory [12,38].

##### Quinoa and Amaranth

Quinoa and amaranth seeds are gluten-free pseudo cereal grains originally from the Andean region in South America. Apart from a high content of dietary fibre and high quality of protein, quinoa and amaranth contain poly-unsaturated fatty acids and an abundance of anti-inflammatory phytochemicals [128]. In a colitis murine model, consumption of quinoa suppressed the dysbiosis of gut microbiota by increasing species richness and diversity and alleviated clinical symptoms [129]. Methanol extracts of the dried leaves and seeds of amaranth were effective against pathogenic bacteria *S. aureus* and *E. coli*, and fungi Fusarium solani and Rhizopus oligosporus [130]. Assessment of the prebiotic potential of quinoa and amaranth through in vitro cultures with human faecal microbiota showed that both quinoa and amaranth stimulated the growth of certain numerically predominant bacterial groups in the human intestinal microbiota. Proliferation of these bacterial groups increased the SCFAs (acetate, propionate and butyrate) which is in line with the decrease in pH. This prebiotic potential could improve or maintain gastrointestinal health through the equilibrium of intestinal microbiota [131].

#### 3.1.8. Oil and Fats

##### Olive Oil

Extra-virgin olive oil is a central component of the Mediterranean diet. Its main fatty acid is oleic acid, which is associated with a lower risk of cardiovascular disease and displays a protective effect in liver dysfunction and gut inflammation. But extra-virgin olive oil is particularly rich in phenolic constituents, that are completely absent in oils derived from seeds or fruits [132]. These phenolic compounds help prevent oxidative damage in colon cells and improve the symptoms of chronic inflammation in IBD [133]. Extra-virgin olive oil attenuates chronic inflammation by inhibiting arachidonic acid and NF-κB signalling pathways. Recently, a cross-over trial in 32 patients demonstrated that consumption of 50 mL of extra virgin olive oil decreased ESR (*p* = 0.03) and CRP (*p* < 0.001) compared to canola oil [134].

Early studies on enteral nutrition demonstrated that olive oil-based diets better reduced inflammation than diets based on seed oils (corn, safflower, sunflower, soybean), suggesting that oleic acid would have better anti-inflammatory properties than linoleic acid [135]. This was confirmed by two experimental studies [136,137]. However, an earlier study by Gassull et al. randomising 62 patients with active CD to polymeric enteral diets with identical macronutrient distribution containing either predominantly oleic acid (79%) or linoleic acid (45%) as a fat source or prednisolone, demonstrated the highest remission rates for steroids [138]. Surprisingly, the enteral diet with high rates of linoleic acid demonstrated higher remission rates (52%) than the diet with high oleic acid (20%). However, in this study synthetic trioleate was used as the oleate source. Therefore, other components of olive oil might enhance the anti-inflammatory capacity. 

In DSS-induced colitis in mice, administration of extra-virgin olive oil reduced weight loss and rectal bleeding and enhanced intestinal permeability [133]. Compared to sunflower oil, extra virgin olive oil-enriched diets showed a reduction of proinflammatory cytokine levels in DSS-induced mice [139]. In HLA-B27 transgenic rats, extra virgin olive oil reduced TNFα gene expression in the colonic mucosa and reduced total blood cholesterol compared to corn oil but did not attenuate intestinal inflammation [140].

An in vitro study showed that the unsaponifiable fraction of extra virgin olive oil stimulates apoptosis and reduces activation of intestinal and blood T cells isolated from IBD patients and impairs the expression of the gut homing receptor integrin β7 on blood T cells from IBD patients [141].

##### Sunflower Oil

Sunflower oil is pressed from the seeds of sunflower and is primarily composed of linoleic acid, a polyunsaturated fat, and oleic acid, a monounsaturated fat [142]. In the Western diet, linoleic acid is the highest consumed PUFA. It is the precursor of arachidonic acid. Therefore, its ubiquitous consumption could lead to changes in arachidonic acid tissue levels. But this relationship could not be proven in a systematic review [143]. Furthermore, sunflower oil is a rich source of vitamin E. Compared to extra virgin olive oil, sunflower oil showed higher disease activity index and demonstrated higher proinflammatory cytokine levels in DSS-induced colitis in mice [139]. 

##### Canola Oil

Canola oil is a vegetable oil pressed from rapeseeds. In DNBS-induced colitis in rats, administration of canola oil, safflower oil and fish oil were compared. Fish oil showed the highest body weight, followed by canola oil. Safflower oil showed the most severe intestinal damage, followed by canola oil. Fish oil showed the least histological damage [43].

##### Coconut Oil

Coconut oil contains 90.5% of saturated fatty acids (SFAs), but only 8.8% of monounsaturated fatty acids (MUFAs) and 0.5% of PUFAs. A prospective study including 412 patients with quiescent UC treated with an aminosalicylate, reported that dietary intake of myristic acid, a SFA commonly found in palm oil, coconut oil and dairy fats was associated with increased risk of a flare (OR 3.01; 95% CI: 1.17–7.74) [30]. Coconut oil could not demonstrate a beneficial effect in DSS-induced colitis in mice [144].

##### Butter

Butter is derived from milk fat. No human studies on butter and IBD could be found. Dietary studies on dairy and milk often reported to worsen the symptoms, but epidemiological studies show conflicting results [23]. As butter contains butyrate, a type of fatty acid that is also produced by gut bacteria and has been demonstrated to inhibit inflammation by enhancing NF-ĸB and feeding colonocytes [145], one might argue that butter is beneficial for patients with IBD. However, a recent systematic review demonstrated that sodium butyrate enemas have limited effect on histological and inflammatory parameters in patients with IBD [146]. Animal studies showed that diets high in saturated milk fat promoted Bacteroidetes and decrease Firmicutes in Il10−/−mice [147]. Furthermore, this diet promoted growth of *Bilophila wadsworthia*, which is often highly abundant in human pathological conditions such as appendicitis and regarded as pro-inflammatory due to its immune activating and sulphite-reducing properties. In Il-10 deficient mice, a high milk fat diet accelerated the onset of colitis promoting taurine conjugation of bile acids, greater availability of luminal sulphur and subsequent expansion of a specific sulphite-reducing opportunistic pathogen, *Bilophila wadsworthia*. 

##### Margarine

High intakes of margarine correlate with increased risk of UC [148,149], but this might also be due to the high content of linoleic acid, an n-6 PUFA highly present in margarine [150], and high amounts of trans fatty acids which were previously found in margarine. On the other hand, margarines lower in n-6 PUFA, but enriched with omega-3 fatty acids or plant sterols might be beneficial. Docosahexaenoic acid (DHA) inhibits dimerisation of Toll-like receptor-4 (TLR-4) [151,152]. A positive effect of plant sterols and stanols on Treg presence in the colon in a DSS model, could only be demonstrated in a low-fat diet [153].

#### 3.1.9. Nuts

Nuts are nutrient-dense foods with rich sources of unsaturated fatty acids, fibre and protein, along with many vitamins (tocopherol, pyridoxine, niacin or folic acid), minerals (Mg, K and Cu) and other phytochemical constituents (stigmasterol, campesterol, resveratrol and catechins) [154]. Pistachios, pecans and walnuts are rich sources of phenolic compounds, including anthocyanins, flavonoids, proanthocyanidins, flavonols, isoflavones, flavanones, stilbenes, phenolic acids and hydrolysable tannins, which are important as antioxidants [155,156]. Nut consumption has a positive influence on health outcomes like hypertension, diabetes, CVD, cancer, other inflammatory conditions and mortality. However, the servings of nuts and seeds was inversely associated with BMI in IBD patients (R-0.049; *p* < 0.05) [44].

##### Pistachio

Among nuts, pistachios have a lower fat (mostly from PUFA and MUFA) and energy content and higher levels of fibre (both soluble and insoluble) and the highest levels of vitamin K, γ-tocopherol, phytosterols, xanthophyll carotenoids, certain minerals (Cu, Fe and Mg), pyridoxine and thiamine [154].

Pistachios have a high antioxidant and anti-inflammatory potential [157], which might be explained by their high γ-tocopherol content [158], or antioxidant effects of Zn and Se. Both minerals have been recognised to be involved in the prevention of CVD and some types of cancer [159,160]. Furthermore, raw pistachios contain about thirteen times more lutein and zeaxanthin than hazelnuts, the next highest nut type [161]. Lutein and zeaxanthin are two xanthophyll carotenoids responsible for giving colour to pistachio nuts and have antioxidant properties. After roasting and steam roasting, a common method to increase their overall safety and palatability, the antioxidant capacity and total phenol content were reduced by 60% and lutein was even more stable in this process compared with other types of carotenoids [162]. 

A randomised, controlled, cross-over trial in 28 hypercholesterolemic adults showed that the consumption of diets containing 10 and 20% of energy from pistachios (32–63 and 63–126 g/day, respectively) increased antioxidant concentrations in serum, such as γ-tocopherol, lutein and β-carotene, whereas it decreased oxidised LDL concentrations compared to a control diet without pistachios [163]. 

In a study conducted in 44 healthy volunteers, equally randomised to a regular nut-free diet group and a pistachio group (in which pistachios accounted for 20% of their daily energy intake) for 3 weeks, the blood antioxidant potential (determined by the production of thiobarbituric acid-reactive substances) increased and malondialdehyde levels decreased in the volunteers consuming pistachios, which implicates decreased oxidative stress [164]. 

Finally, in a prospective study, 32 healthy young men followed the Mediterranean diet for 4 weeks. During the next 4 weeks, the Mediterranean diet was supplemented with pistachios by replacing the monounsaturated fat content constituting approximately 20% of daily energy intake [165]. The pistachio diet significantly improved endothelium-dependent vasodilation, decreased serum IL-6, total oxidant status, lipid hydroperoxide and malondialdehyde and increased superoxide dismutase, whereas there was no significant change in C-reactive protein and TNF-α levels. Taken together, these results provide evidence of anti-inflammatory effects of pistachios.

##### Walnuts

A randomised crossover trial in 18 healthy individuals showed that 42 g of walnut consumption resulted in a 49–160% higher relative abundance of Faecalibacterium, Clostridium, Dialister and Roseburia and 16–38% lower relative abundances of Ruminococcus, Dorea, Oscillospira and Bifidobacterium (*p* < 0.05) compared with after the control period [166]. Furthermore, it reduced microbially derived, proinflammatory secondary bile acids and LDL cholesterol. In another crossover RCT in 194 healthy individuals, 43 g of walnut consumption for eight weeks significantly increased the abundance of Ruminococcaceae and Bifidobacteria (*p* < 0.02) while Clostridium sp. cluster XIVa species (*Blautia; Anaerostipes*) decreased significantly (*p* < 0.05) [167]. Thus, walnut consumption significantly enhanced probiotic- and butyric acid-producing species in the microbiota of healthy individuals.

In a rat study, administration of 6 or 9% walnut diet, significantly suppressed activation of P38-Mitogen-activated protein kinase (MAPK) and NF-kB in brain tissues, both being proinflammatory molecules [168].

##### Almonds

A randomised crossover study in 18 healthy individual showed that daily almond consumption of 42 g increased the relative abundance of Clostridium clusters IV and XIVa, including Roseburia, Clostridium, Lachnospira compared to the control diet period without almonds [169]. Furthermore, comparisons between whole, whole roasted, roasted chopped and butter almonds and controls revealed that consumption of roasted chopped almonds increased the relative abundances of Roseburia, Lachnospira and Oscillospira. Whole roasted almonds also increased the relative abundance of Lachnospira. Consumption of almond butter showed no differences in the microbiota compared to the control group.

#### 3.1.10. Cocoa and Chocolate

Cocoa beans are the fermented ground seeds from the cacao tree (Theobroma tree) which are used to make cocoa liquor. This paste consists of non-fat cocoa mass and cocoa butter [170]. Cocoa powder is obtained by removing some of the cocoa butter from the liquor. Chocolate is a combination of cocoa liquor with cocoa butter and sugar. 

Cocoa and dark chocolate possess polyphenols which have antioxidant and anti-inflammatory effects. The polyphenols in cocoa powder are mainly composed of catechins, anthocyanins and proanthocyanidins [171]. Cocoa polyphenols activate important signalling pathways such as TLR 4/NF-κB/signal transducer and activator of transcription (STAT). Furthermore, polyphenols modulate intestinal microbiota, thus leading to the growth of bacteria triggering an anti-inflammatory pathway in the host [172]. 

Cocoa butter also contains both monounsaturated (mostly oleic acid) and saturated (mainly palmitic and stearic) fatty acids [90,173]. 

The fibre present in cocoa beans has shown to improve the ratio between low and high density lipoprotein [174], and reduce the risk of type 2 diabetes [175].

Cocoa and chocolate contain significant amounts of magnesium, copper, potassium and iron [176]. Magnesium, copper and potassium reduce the risk of cardiovascular disease [177,178,179], while iron, mainly present in dark chocolate, contributes to preventing anaemia [180].

Chocolate consumption was positively associated with the risk of developing CD (OR 2.5; 95% CI: 1.8–3.5) and UC (OR 2.5; 95% CI: 1.8–3.5) [72]. However, in healthy volunteers, cocoa consumption reduced NF-κB in peripheral blood mononuclear cells (PBMCs), thus suggesting reduced release of pro-inflammatory cytokines [181]. Shapiro et al. even concluded that addition of polyphenols to artificial enteral nutrition in patients with inflammatory bowel disease might prevent or improve their inflammatory status [182].

Research in murine models of dextran sulphate sodium (DSS)-induced colitis showed that administration of cocoa FGM-derived polyphenols led to significant abrogation of intestinal length reduction, accompanied by a significant drop of TNF-α and IL-1β in the inflamed colon [183]. Similarly, administering a 5% cocoa diet to these murine models demonstrated anti-inflammatory potential by down-regulated serum TNF-α, colon inducible nitric oxide synthase activity and decreased colon cell infiltration [184]. Furthermore, administration of cocoa polyphenols reduced symptoms paralleled with reduced neutrophil infiltration, NO generation, expression of COX-2 and STAT-1 and STAT-3 as well as reduction of IL-1β, IL-6 and TNF-α from peritoneal macrophages [185]. Reduction of these biomarkers was associated with improvement of colitis. Studying the effect of cocoa and dark chocolate on intestinal inflammation, in vitro treatment of Caco-2 cells with cocoa polyphenols showed induction of prostaglandin E2 synthesis via cyclooxygenase (COX)-1 effect, which may be involved in maintaining mucosal integrity of the gut [186].

#### 3.1.11. Coffee

Coffee contains caffeine and chlorogenic acid, a phenolic acid [64].

A meta-analysis evaluating six studies showed that coffee consumption tended to be inversely associated with UC risk (RR 0.58; 95% CI: 0.33–1.05). This relation became significant without adjustment for smoking [187]. Furthermore, decaffeinated coffee demonstrated better endoscopic results in 81 UC patients in all stages of the disease than in patients using caffeine-containing coffee [188].

In DSS-induced colitis in mice, a caffeine dose of 2.5 mmol/L, equivalent to the concentration of caffeine in 2–3 cups of coffee, showed anti-inflammatory effects by down-regulating the expression of chitinase3-like 1 (YKL-40) in the colon [189].

#### 3.1.12. (Green)Tea

A meta-analysis of three studies showed a pooled RR for the highest versus the lowest tea consumption for UC risk of 0.67 (95% CI: 0.58–0.83) [187]. Green tea is known to increase beneficial bacteria [7,190].

A review on the effects of green tea polyphenols in UC and CD included ten studies. From one clinical and nine mouse model studies, green tea polyphenols were shown to play a role as antioxidant and have anti-inflammatory effects by downregulation of NF-kB, TNF-α, IL-1β and other cytokines, and due to their prebiotic effects establishing a healthy gut microbiota or acting on regulation of claudin or occludin [191].

#### 3.1.13. Sweetened Beverages

A meta-analysis of five studies showed a pooled RR for the highest versus the lowest soft drinks consumption and UC risk of 1.69 (95% CI: 1.24–2.30, I^2^ = 12.9%, *p*_heterogeneity_= 0.332) [187]. After excluding cases occurring within the first two years after the dietary assessment, a positive association was found between a ‘high sugar and soft drinks’ pattern and UC risk in the EPIC cohort study (an adjusted incidence rate ratio for the fifth versus first quintile of 1.68 (95% CI: 1.00–2.82; *p*_trend =_ 0.02) [192]. Cola drinks were positively associated with the risk of developing CD (OR 2.2; 95% CI: 1.5–3.1) as well as UC (OR 1.6; 95% CI: 1.1–2.3) [72]. However, recently in a prospective cohort study with 143 incident cases of CD and 349 incident cases of UC, Khalili et al. found no significant multivariable-adjusted HRs for 1 or more servings of sweetened beverages per day (for CD: HR 1.02 (95% CI: 0.60–1.73; for UC HR 1.14 (95% CI: 0.83–1.57)) compared to participants who reported no consumption of sweetened beverages [193].

Compared to CD patients with inactive disease, those with active disease consumed more sweetened beverages (OR 1.29; 95% CI: 1.09–1.53) and soda (OR 1.51; 95% CI: 1.29–1.78) [23]. IBD patients with active disease consumed significantly more sports drinks (mean, 2.3 bottles vs. 1 bottle) and sweetened beverages (mean 5.1 vs. 2.6 cups) compared with those with inactive disease per week [26]. CD patients with an ostomy consumed more sweetened beverages compared to those without an ostomy (OR 2.14; 95% CI: 1.02–1.03) [23].

Surprisingly, very recently, in vitro experiments demonstrated that energy drink supplementation enhanced the epithelial barrier defects in a dose-dependent, whereas in vivo experiments in DSS-induced colitis, energy drink decreased clinical symptoms and epithelial barrier permeability [194]. 

#### 3.1.14. Alcohol

A meta-analysis evaluating nine studies found no association between alcohol consumption and UC risk (RR for highest vs. lowest consumption level: 0.95; 95% CI: 0.65–1.39) [187].

##### Wine

Wine is an integral part of the Mediterranean diet which is generally considered to be beneficial for health. Moderate wine consumption is associated with a reduced incidence of oxidative stress-related degenerative diseases [195]. However, evidence is still limited and attention should be paid in discriminating between the effects of alcohol and those of other wine components in human intestinal diseases [196].

On the one hand, wine has anti-oxidant properties, mainly attributed to a large variety of phenolic compounds, which are mainly present in red wine [197]. These phenolics act as both free radical scavengers and modulators of specific inflammation-related genes involved in cellular redox signalling [196]. Phenolics may also reduce harmful bacteria and stimulate beneficial gut bacteria, especially *Bifidobacteria* and *Lactobacteria* [198,199]. In an RCT cross-over study, healthy volunteers consumed a placebo and either a grape juice extract or a mixture of grape juice and wine polyphenol extract over a 4-week period. The grape juice extract did not change the faecal metabolic profiles compared to the placebo, whereas the mixture with grape juice with wine polyphenols decreased in isobutyrate, suggesting that polyphenols can modulate the microbial ecology of the gut [200]. Besides, ascorbic acid and sulphur dioxide, commonly added to white wine, may also enhance the antioxidant properties of white wine [201]. 

On the other hand, wine contains constituents which might be associated with oxidative damage such as alcohol (ethanol) which is produced during grape fermentation, and sulphite.

Moderate alcohol consumption often comprises about 15 g/day for women and 30 g/day for men, corresponding to 1–3 glasses of wine. Whereas high alcohol intake demonstrates detrimental health effect, low alcohol consumption is mostly associated with reduced risk of cardiovascular disease. However, human studies on the effects of wine, alcohol and phenolics on intestinal diseases are scarce and show conflicting results [202]. 

A study in which 8 patients with inactive UC and 6 patients with inactive CD consumed 1–3 glasses of red wine a day showed that alcohol intake was on the one hand associated with decreased stool calprotectin (*p* = 0.001) but on the other hand with increased small bowel permeability (*p* = 0.028). This might suggest that drinking red wine on a regular basis might increase the risk for flares [203]. Inclusion of patients with sustained large alcohol consumption contributes to gut motility disorders. Furthermore, sustained high alcohol concentrations instigates constant oxidative and inflammatory reactions causing serious mucosal damage [204,205].

In a prospective study, 81 UC patients at all stages of the disease completed a 7-day diet diary and were examined by sigmoidoscopy in order to elucidate any association between individual dietary items and disease activity [188]. This study revealed that sulphite-containing alcoholic beverages such as wines and beers, were associated with higher UC disease activity, but spirits were not. This indicates a more pronounced role for sulphite than alcohol in the disease activity. Besides, hydrogen sulphide present in wine, has been hypothesised to induce pro-inflammatory and genotoxic effects, for its interfering activity in sulphate-reducing bacteria and colonocyte metabolism in butyrate oxidation [206]. 

A crossover study including 20 patients with CD in remission, who randomly consumed red and white wine, Smirnoff Ice, Elephant Beer and pure ethanol showed no differences in alcohol absorption compared to the healthy subjects (*n* = 10). However, CD patients reported significantly more abdominal pain manifestations after consumption of Smirnoff Ice and Elephant beer with high glucose concentration compared to consumption of red and white wine or pure ethanol. This suggests that fermentation might be responsible for intestinal bacteria overgrowth, thus playing a role in abdominal pain and intestinal damage in IBD [207].

In in vivo models ex vivo organoids generated from jejunum and colon from mice, alcohol leads to a reduced ratio of butyrate/total short chain fatty acids (SCFA) in stool and increased colonic hyperpermeability [208].

#### 3.1.15. Yeast

*Saccharomyces boulardii* is a non-pathogenic strain of yeast [209]. Previously, Hunter described that exclusion of cereals, dairy products and yeast prolonged remission for CD patients [210]. A study of Triggs et al. reported that yeast-based breads and cereals showed more GI symptoms compared to less processed cereals and sourdough breads were best tolerated [54].

A recent review of *Saccharomyces boulardii* as a treatment option in IBD found only limited evidence [211]. Three of the 16 included clinical trials showed a beneficial effect on IBD. Two of them were conducted in CD patients who were in remission [212,213], while another study included UC patients with active disease [214]. Only small populations were included, and the treatment effect was measured in different ways. A fourth clinical trial in CD patients in remission showed *S. boulardii* did not prevent relapse [215]. In the exclusion diet for CD-patients with flares (CDED), consumption of yeast is not allowed [12,38].

#### 3.1.16. Sugar

Saccharose was associated with CD risk (pooled RR 1.088; 95% CI: 1.020–1.160, I^2^ = 0.0%; *p*_heterogeneity_ = 0.395 per 10 g increment/day) [216]. In animal models, high saccharose consumption induced tissue inflammation [65,217,218]. Furthermore, high saccharose intake could induce endoplasmic reticulum stress which was also associated with CD risk [219,220].

High sugar intake was associated with higher odds for CD (OR 3.50; 95% CI: 1.73–7.07) but not in UC [19]. In the European Prospective Investigation into Cancer and Nutrition (EPIC) study a positive association between a pattern ‘high in sugar and soft drinks’ and UC risk (1.68; 95% CI: 1.00–2.82; *p*_trend_ < 0.05) was described, but only if they had low vegetable intakes [192].

Another study found that saccharose intake was positively associated with UC risk (RR for per 10 g increment/day: 1.098; 95% CI: 1.024–1.177) [221].

The SCD eliminates all sugars (except for honey) [9].

#### 3.1.17. Honey

Consumption of honey contributes many beneficial effects to CD patients [54]. Honey is the most commonly used complementary and alternative medicine therapy reported by Saudi IBD patients (62%) [222]. Acacia honey and citrus fruit contain a bioactive constituent flavonoid called acacetin. Oral administration of acacetin increased the production of iNOS, COX-2, IL-6, TNF-α and IL-1β in mice with DSS-induced colitis [223]. Besides, in vitro acacetin led to a significant decrease of macrophage infiltration, which might also enhance its therapeutic effect in IBD.

The SCD eliminates all sugars but allows honey [9]. 

#### 3.1.18. Salt

The Western diet with many processed foods is a high provider of dietary sodium (sodium chloride). Increased dietary salt intake has been associated with the development of autoimmune diseases. Only one human study reported data concerning salt consumption and IBD. In a US cohort from 194,711 women, dietary intake of potassium (*p*_trend_ = 0.005) but not sodium (*p*_trend_ = 0.440) was inversely associated with CD risk [224]. They did not find any significant association between both dietary potassium and sodium and risk of UC. But the role of high-salt diets on the course of IBD still needs to be studied. In healthy volunteers, a high dietary salt intake was associated with enhanced blood monocyte counts and pro-inflammatory cytokines such as IL-6, IL-17 and IL-23 but lower anti-inflammatory IL-10 [225,226,227]. 

Tubbs et al. reviewed literature on salt and colitis in animals [228]. In experimental murine colitis models, exposure of intestinal cells to a high-NaCl diet induced inflammatory cytokine production such as IL-23 and IL-17 in normal intestinal lamina propria and contributed to the exacerbation of experimental colitis [229]. Inducing IL-23-IL-23R-mediated induction of pathogenic TH17 cells plays a critical role in development of IBD [230,231,232]. Besides stimulation of the intestinal TH17 response, dietary salt also inhibits the suppressive functions of Foxp3+ Treg cells [233,234]. Furthermore, dietary salt may activate macrophages creating a more inflammatory environment [235]. High salt levels inhibit alternative activation of macrophages causing attenuation of tissue inflammation [236]. Likewise, sodium attenuates the Treg function by stimulating IFNγ production in these cells. 

Furthermore, high dietary salt exacerbates murine experimental colitis and has a deleterious impact on faecal microbiota by decreasing Lactobacillus levels and butyrate production. This detrimental effect of high dietary salt was not observed in germ-free mice [237]. Surprisingly, Tubbs et al. found that sodium accumulated in the colons of mice with a high dietary salt intake, suggesting a direct effect of salt within the colon [228]. These findings suggest that high consumption of dietary salt might be detrimental for the development and course of gut inflammation in IBD patients. However, causal relationships still need to be proven. Surprisingly, countries like China with the highest salt intakes, have lower rates of IBD compared to the Western countries [238,239]. 

#### 3.1.19. Herbs and Spices

Herbal products are widely used by IBD patients, although a lot is based on ancient ideas or beliefs. Many dried herbs such as thyme, oregano and basil contain many polyphenols, and ginger and cumin have an anti-inflammatory effect [240].

A systematic review on herbs in patients with IBD found several trials [241]. First in 40 patients with active CD treated with prednisolone, 500 mg three times daily of wormwood (*Artemisia absinthium*) significantly improved Crohn’s Disease Activity Index (CDAI) after 8 weeks compared to placebo. After 20 weeks 90% of the wormwood group was steroid-free, compared to 20% in the placebo group [242]. In another study, after 20 weeks of wormwood 750 mg three times daily, in 20 CD patients with active disease, 80% demonstrated a clinical response and 60% were in remission, as opposed to 20% and 0%, respectively in the placebo group [243].

Furthermore, a randomised study in 102 patients with active CD comparing *Boswellia serrata* with high-dose mesalamine, demonstrated no significant difference in CDAI [244]. Although the follow-up was only 8 weeks, it did demonstrate some efficacy in view of noninferiority to high dose mesalamine. In patients with inactive disease *Boswellia serrata* did not show any difference with a placebo [245].

In CD patients with inactive disease *Tripterygium wilfordii*, a vine-like plant, demonstrated less clinical and endoscopic recurrence compared to mesalamine [246], but similar clinical occurrence rates and higher endoscopic recurrence compared to azathioprine [247]. High dose *T wilfordii* demonstrated lower relapse rates (10%) compared to low-dose *T wilfordii* (22%, *p* = 0.047) and mesalamine (29%, *p* = 0.006). However, this was related with more side effects (30%, 28% and 14% respectively, *p* < 0.05) [248].

In UC, many different types of herbs have been tested [241]. Wheat grass juice is produced from *Triticum aestivum,* which is a cultivated wheat containing flavonoids, phenols and chlorophyll. In an RCT in 23 patients with active distal UC, consumption of 100 mL wheat grass juice demonstrated lower rectal bleeding (*p* = 0.025), disease activity scores (*p* = 0.019) and better physician’s global assessment scores (*p* = 0.031) compared to placebo, but sigmoidoscopic scores did not differ [249]. *Andrographis paniculate* (Indian Echinacea) is an herbal mixture for respiratory infections and diarrhoea. This herbal mixture was compared to mesalamine in patients with moderately active UC and demonstrated noninferiority on a composite score of clinal, endoscopic and histological outcomes [250]. Furthermore, in patients with mild to moderately active UC, Indian Echinacea demonstrated higher mucosal healing rates (50%) compared to placebo (33%), but similar remission rates [251]. In two clinical trials, *Boswellia serrata* did not demonstrate significant differences in symptom severity, remission rates or safety compared with sulfasalazine [252,253].

Lastly, curcumin was tested. Curcumin is a phytochemical present in the rhizomes of the herb ‘Curcuma longa’. In Asia, it has been used for centuries, both in traditional medicine and in cooking as turmeric is a spice that gives food an exotic natural yellow colour [254]. Curcumin was tested orally in an RCT including patients with mild to moderately active UC. Compared to placebo, curcumin supplementation demonstrated better clinical response, and higher clinical and endoscopic remission rates [255]. However, a study comparing an enema of curcumin to placebo in the UC patients with mild to moderate UC found no significant differences [256]. In patients with quiescent UC, curcumin vs. sulfasalazine or mesalamine, curcumin showed lower relapse rates and enhanced clinical activity and endoscopic scores after 6 months, but these differences were not sustained after 12 months [257]. A Cochrane review and a later review both concluded that curcumin may be a safe and effective therapy for maintenance of remission in quiescent UC when used in combination with mesalamine or sulfasalazine but more research was needed [258,259].

In mice, lemon grass (*Cymbopogon citratus*) enhanced ileitis by decreasing lymphocyte recruitment to the inflamed intestine [260].

Furthermore, herbal therapies can have direct or indirect side effects. In a direct way toxicity from herbal therapies may lead to fatal liver and renal failure [261,262]. Some herbal treatments included mercury, arsenic or lead. Probably also due to lack of post marketing surveillance, side effects of herbal products might have been overlooked [241]. Probably the indirect effect of initially consulting alternative practitioners leading to a delay or wrong diagnose might even be more harmful.

#### 3.1.20. Processed Foods and Food Additives

Apart from added sugar and salt, processed foods of contain food additives such as emulsifiers, thickeners or microparticles. Emulsifiers and thickeners are added to many processed foods such as breads and other baked goods, fat spreads, mayonnaises and salad dressings, ice creams and other dairy desserts, confectionery and beverages [263]. Synthetic emulsifiers, such as carboxymethylcellulose and polysorbate-80 are commonly added to a variety of processed foods to enhance the texture or extend shelf-life. Most emulsifiers are destroyed by digestive processes in the proximal small intestine, but thickening agents such as carboxymethylcellulose and carrageenan potentially have detrimental effects within the whole gastrointestinal tract [263]. In mice studies, dietary carboxymethylcellulose and polysorbate-80 led to alterations in microbiota composition, microbiota encroachment into the mucus and development of chronic inflammation [264]. Dietary methylcellulose fibre exacerbated colitis in specific pathogen-free mice [265]; similar results were found for carboxymethylcellulose [266]. 

Also food additives such as aluminium, titanium dioxide (TiO_2_) and microparticles may promote intestinal inflammation and attribute to the disease progression or relapse in individuals with IBD [267]. Microparticles are used as anticaking agents or food colorants. In murine models of colitis, accumulation of these microparticles have been demonstrated in Peyer’s patches [268]. Although, detrimental effects could not be demonstrated in a single-blind, randomised, multi-centre, placebo controlled trial in 83 patients active CD with a 36 week follow-up [269].

Carrageenans (E407) are extracted from red and purple seaweeds, and commonly used as a thickening or emulsifying food additive to enhance the texture of dairy products such as chocolate milk, ice cream, cottage cheese, sour cream and yogurt, soy milk, almond milk, mayonnaise, but also infant formula and processed meats [270]. Studies in different animals consistently report that carrageenan and carboxymethylcelluloses induce histological changes that are typical of IBD, change the microbiome, increase intestinal epithelial permeability, inhibit proteins that provide protection against microorganisms and trigger pro-inflammatory cytokines [271]. These findings are confirmed in recent in vitro studies of human epithelial cells and the human microbiome. Carrageenan and carboxymethylcellulose may trigger or magnify an inflammatory response in the human intestine [271].

Maltodextrin is a ubiquitous food additive used as a thickener or sweetener. 60% of all packaged food items in grocery stores have ‘maltodextrin’ or ‘modified (corn, wheat, etc.,) starch’ included in their ingredients list [272]. Higher maltodextrin consumption coincided with a substantial increased CD incidence [273]. In ileal CD patients, higher levels of *E. coli* and adherent-invasive *E. coli* strains were observed, suggesting a role for *E. coli* in disease pathogenesis [274,275,276]. Maltodextrin promotes multiple *E. coli* strains, including adherent-invasive *E. coli*, stimulating cellular adhesion and biofilm formation on the gut epithelium of CD patients [276]. Compared to samples of colonic CD patients and healthy controls, samples of ileal CD patients demonstrated increased levels and prevalence of a specific maltodextrin-binding component [277]. These findings suggest that maltodextrin consumption may stimulate *E. coli* colonisation and contribute to CD susceptibility [272]. Studies in preterm pigs showed maltodextrin consumption stimulated the expansion of ileal *E. coli* [278] and induced necrotising enterocolitis, but not in fully developed pigs [274]. In mice, diets enriched with maltodextrin exacerbated inflammation by increasing endoplasmic reticulum stress in epithelial cells of the gut reducing the mucus production and exacerbating intestinal inflammation [279].

The SCD eliminates most processed foods [9]. Processed foods are also not allowed in the CDED, IBD-AID and low FODMAP diet [8,12,14].

#### 3.1.21. Miscellaneous

Chewing gum was positively associated with the risk of developing CD (OR: 1.5; 95% CI: 1.1–2.1) [72].

### 3.2. Comparison between Different IBD Diets and the GrAID Diet on Food Groups

In Table 1 an overview is presented on different food groups in several known IBD diets compared to the GrAID. 

Regarding meat products, only in the CDED red meat is prohibited, but lean meat is allowed and consumption of at least 150–200 g chicken daily is mandatory in the CDED [12,38]. In Mediterranean diet red and processed meat are not used very often [280]. In the IBD-AID lean meats, poultry, fish and omega-3 eggs are used [8], whereas in FODMAP diet there are no restrictions to meat consumption [14]. In the SCD fresh meats are allowed, but processed meats should be avoided [10]. In the GrAID, the use of processed meat and red meat is limited to once a week; the use of lean and unprocessed meat are allowed. Chicken and eggs can also be used as these are an important protein source. 

In the IBD-AID use of aged cheeses, fresh cultured yogurt and kefir are allowed, but lactose-rich dairy products should be avoided [8]. In the GrAID, dairy products should be consumed as pure as possible, such as plain milk, buttermilk of fermented milk such as yogurt of kefir. Fermented milk and hard cheeses also have the advantage that they contain less to no lactose, which might be a problem in some IBD patients. 

The FODMAP diet is highly restrictive on certain fruits and vegetables [14]. In the IBD-AID only specific fruits and vegetables should be used [8]. Specific products with inulin such as bananas are recommended. One apple and two bananas a day are even mandatory in the CDED diet [12,38]. Some legumes are permitted such as lentils and split peas, but chickpeas and soybeans are not allowed. In the GrAID fruits, vegetables and legumes can be used unlimited. Some people experiencing adverse effects of fruits or vegetables such as corn, could leave these out. Besides, potatoes and rice can be used, and cooked and cooled consumption is recommended because this increases resistant starch. 

The CDED is an exclusion diet avoiding gluten, dairy products, gluten-free baked foods and breads [12,38]. In the SCD no cereal grains or quinoa are permitted. In the FODMAP diet rice and oats are allowed, whereas wheat and rye should be avoided [14]. Although bread and wheat may raise some concern, but as bread and other wheat products are so widely used, restricting these products would limit a lot of IBD patients. Therefore, the GrAID recommends high fibre breads and cereals without food additives. 

The GrAID prefers olive oil or fats containing N-3 fatty acids over sunflower oil or saturated fats. Nuts are allowed in the IBD-AID, the SCD and are often eaten in the Mediterranean diet [8,10,280]. The GrAID recommends consumption of nuts. In the GrAID and IBD-AID limited use of chocolate is allowed, whereas it should be avoided in the SCD [10]. In the CDED consumption of yeast is not allowed [12,38], but it is allowed in the GrAID.

Beverages: In the GrAID coffee and tea can be used unlimited, but sweetened beverages and alcohol should be avoided. Wine is a key feature of the Mediterranean diet [280]. In the SCD wine is allowed, but beer, instant tea and coffee are prohibited [10]. In the CD-TREAT, a diet mimicking E.E.N., alcohol is also prohibited [13].

The SCD allows saccharin and honey, and moderate use of sorbitol and xylitol. In the FODMAP diet honey is prohibited [14], whereas in the IBD-AID it is recommended [8]. In the GrAID, adding sugar should be avoided as much as possible. When necessary, use of honey is preferred over sugar. 

In the SCD canned fruits and vegetables are not permitted due to possible added sugars and starches [10]. In the CDED diet all packaged foods, processed meats and products containing emulsifiers are not allowed [12,38]. As processed foods often contain added sugar, large amounts of salt and food additives, canned and processed foods should be avoided in the GrAID. Furthermore, adding salt should be kept to a minimum. 

## 4. Discussion

Reviewing literature studying the effects of food and food groups on the onset and course of IBD demonstrated conflicting results on several foods and not all effects on IBD of every single food can yet be proven. However, since patients with IBD are keen to know what they can eat or should avoid in order to stay in remission. This review provides them with a good overview. Based on this evidence, the Groningen anti-inflammatory diet (GrAID) was designed and is about to be tested in an RCT.

The GrAID was designed to provide IBD patients with the best evidence there is on food and food group level to stay in remission without unnecessary limiting patients. The GrAID provides IBD patients a list of foods, that have shown to be safe or beneficial in the course of IBD or should be avoided because they show detrimental effects. But there is also a group of foods that have both beneficial and detrimental effects. Since dietary research is complex, we do not claim that this diet is perfect and future research might lead to attenuation of the GrAID diet. By gathering the best evidence there is, we try to create some guidance for those who are willing to prevent flares. 

Although a systematic review revealed that about 35% of patients with quiescent IBD meet the Rome criteria for a functional bowel disorder, this review was not focussed on reducing gastro-intestinal symptoms such as diarrhoea, bloating, flatulence and abdominal pain [281]. The FODMAP diet was originally designed for patients with IBS and may also alleviate symptoms in patients with IBD [282]. In a non-blinded RCT in patients with quiescent IBD and coexisting IBS-like symptoms (Rome III), compared to patients on a habitual diet (*n* = 45), patients on a low FODMAP diet (*n* = 44) showed greater symptom (*p* = 0.02) and QOL (*p* < 0.010 improvements [283]. However, the low FODMAP diet failed to demonstrate anti-inflammatory properties in patients with quiescent IBD and even reduced their faecal abundance of gut microbes [14,15]. However, every person is different and not all patients should avoid the same foods to alleviate the same symptoms, therefore professional dietary guidance is needed to test which foods are causing the GI symptoms and eliminated foods should be properly reintroduced to avoid nutritional deficiencies [284].

There is some conflicting evidence on the use of wheat. Wheat should be avoided in the IBD-AID, SCD, CDED, CD-TREAT and low-FODMAP diet. On the one hand, wheat might trigger GI symptoms [55], and also contains gluten (gliadin) which may aggravate GI symptoms independent of celiac disease [108,109]. About 25% of the IBD report gluten sensitivity [110,111]. However, since this self-reported gluten sensitivity was strongly associated with having had a recent flare (OR 7.45; 95% CI 1.63–34.14; *p* = 0.01). This suggests that gluten sensitivity might be a transient phenomenon in some patients [111]. In an American study, 8% of IBD patients actively used a gluten-free diet to control symptoms, which was associated with improvement of fatigue (*p* < 0.03) [112]. On the other hand, a case-control study demonstrated that daily intake of whole meal bread significantly reduced the odds for CD (OR 0.26; 95% CI: 0.15–0.48) and UC (OR 0.42; 95% CI: 0.26–0.70) [19]. Wheat bran was well tolerated in IBD patients [103]. Furthermore, it protects against colon cancer, a type of cancer, which often appears in IBD patients [104,105]. Besides, the Mediterranean diet, which was associated with reduced risk of CD, also contains wheat [280]. Although bread and wheat may raise some concern, since bread and other wheat products are widely used, restricting these products in IBD patients without proven gluten sensitivity would limit them too much. Therefore, the GrAID recommends high fibre breads and cereals without food additives. 

Nevertheless, studies on dietary effects in patients with IBD are limited. The majority of studies are bound to recall bias [17] and methodological issues [17,18,19]. Besides, accurately capturing dietary intake comes with difficulties since there are potential interactions between nutrients and because it might be affected by the physical form of foods [16]. Therefore, it is not surprising that unambiguous dietary advice has not been developed yet [285,286]. 

Although literature was not reviewed systematically and not on all foods or food groups sufficient studies of good quality were available, this study does present a comprehensive overview of all current best evidence on food and food groups. This will help patients with IBD to select those foods that might be beneficial for the course of the disease and avoid detrimental foods that might cause relapse of the disease. 

## 5. Conclusions

This comprehensive review presents the current state of evidence on food and food groups in the treatment of IBD. Although it is not focussed on GI complaints, it will certainly provide IBD patients with the best evidence on which foods they should use or avoid in order to prevent flares. Based on this evidence, the GrAID was designed which will soon be tested in IBD patients.

## Figures and Tables

**Table 1 nutrients-13-01067-t001:** Overview of GrAID compared to diets being tested in inflammatory bowel disease with mandatory, allowed, limited and not allowed food (groups).

Food (Groups)	GrAID	Mediterranean	IBD-AID	SCD	CDED	CD-TREAT	Low FODMAP	LEGEND:	
Red meat									Mandatory
Lean meat									Allowed
Chicken									Not mentioned
Eggs									Limited
Fish									Not allowed
Dairy products									
Fruit									
Apple									
Banana									
Vegetables									
Legumes									
Corn									
Potatoes									
Wheat									
Olive oil									
Nuts									
Cacao/chocolate									
Coffee									
Green tea									
Sweetened beverages									
Alcohol									
Yeast									
Added sugar									
Refined sugar									
Honey									
Canned food									
Processed foods									

Abbreviations: GrAID, Groningen anti-inflammatory diet; IBD-AID, anti-inflammatory diet; SCD, specific carbohydrate diet; CDED, Crohn’s disease exclusion diet; CD-TREAT, Crohn disease treatment-with-eating diet; Low FODMAP, low-fermentable oligosaccharide, disaccharide, monosaccharide and polyols diet.

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
