# Peer review of "Food and Food Groups in Inflammatory Bowel Disease (IBD): The Design of the Groningen Anti-Inflammatory Diet (GrAID)"

_nutrients, 2021, doi:10.3390/nu13041067_

Round 1
Reviewer 1 Report
- “partly because steroid therapy has adverse effects in children”
Not only in children…
- “several other (partial) whole food diets in IBD”
Why whole?
- Too many acronyms
- “it will take many decades until every single component and food item 73 will be proven beneficial or detrimental”
Will this be true for all patients? For all IBDs?
- Use oxford comma in the whole text
- Define the meaning of acronym GrAID in the full text the first time you used it
- “Furthermore, meat contains saturated fat. Meta-analyses of prospective epidemiologic studies showed that there is no significant evidence for concluding that dietary saturated fat is associated with an increased risk of cardiovascular disease [27,28]. However, recent analyses of two prospective cohort studies in healthy US people showed that an increase in total red meat consumption of at least half a serving per day, especially processed meat, was associated with higher mortality rates [29]”
This part is out of topic
- Don’t use “Crohn’s patients” but patients with CD
- “Soy milk, although not strictly a dairy product, 216 consistently appeared better tolerated than milk products” So, why did you cite soy milk among dairy products?
Reviewer 2 Report
This study reviewed systematically almost all foods or food groups and their effect on IBD. This will help patients with IBD to select those foods that might be beneficial for the course of the disease and avoid detrimental foods that might cause relapse of the disease. As mentioned, an RCT is needed to verify the effect of the proposed diet. But an overview of a huge number of literary sources already provides the basis for dietary recommendations.
Reviewer 3 Report
This is a well-presented comprehensive review of common foods that IBD patients eat as part of their dietary treatment to ideally reduce symptoms and flares. The GrAID designed based on this evidence and still to be tested in an RCT includes most of those foods to some extent compared to the other IBD diets and those not allowed make sense and can easily be avoided.
The authors might want to summarise in the discussion that every person is different, and no diet fits all. In general, patients should focus on quality foods and check food labels to avoid eating food with unnecessary ingredients like food additives that can lead to gut irritations (e.g. choose pure cream without thickener, grass-fed dairy over grain-fed dairy (more omega-6) and organic where possible.
People with colon inflammation likely have allergies, with casein being one of the top allergens. Under the oils section 3.1.9, authors haven’t mentioned butter/ghee as they contain less of the dairy proteins casein and lactose. Ghee contains butyrate, a type of fatty acid that has been shown to inhibit inflammation in some test-tube studies (Segain et al 2000). Butyrate provides energy for the cells in the colon, helps support gut barrier function and fights off inflammation (Riviere et al 2016). I would include evidence to avoid margarine as it increases the risk of IBD (Maconi et al 2010), it’s just a synthetic product with trans fats.
Segain JP, Raingeard de la Blétière D, Bourreille A, Leray V, Gervois N, Rosales C, Ferrier L, Bonnet C, Blottière HM, Galmiche JP. Butyrate inhibits inflammatory responses through NFkappaB inhibition: implications for Crohn's disease. Gut. 2000 Sep;47(3):397-403. doi: 10.1136/gut.47.3.397. PMID: 10940278; PMCID: PMC1728045.
Rivière A, Selak M, Lantin D, Leroy F, De Vuyst L. Bifidobacteria and Butyrate-Producing Colon Bacteria: Importance and Strategies for Their Stimulation in the Human Gut. Front Microbiol. 2016 Jun 28;7:979. doi: 10.3389/fmicb.2016.00979. PMID: 27446020; PMCID: PMC4923077.
Maconi G, Ardizzone S, Cucino C, Bezzio C, Russo AG, Bianchi Porro G. Pre-illness changes in dietary habits and diet as a risk factor for inflammatory bowel disease: a case-control study. World J Gastroenterol. 2010 Sep 14;16(34):4297-304. doi: 10.3748/wjg.v16.i34.4297. PMID: 20818813; PMCID: PMC2937110.
Authors pointed to the fact that vegetable oils are problematic. They are made from grain/seeds/beans like soy, corn, cotton seed, canola and safflower and all those are very damaging for the inflamed colon due to high omega-6 content and the way they have been processed (extracted with high heat, pressure and chemical solvents as don’t have much fat to start with, no nutrients left in these oils just empty calories). Exception is virgin coconut oil and extra-virgin olive oil as healthy alternatives, higher in fat and better to process.
Shilling M, Matt L, Rubin E, Visitacion MP, Haller NA, Grey SF, Woolverton CJ. Antimicrobial effects of virgin coconut oil and its medium-chain fatty acids on Clostridium difficile. J Med Food. 2013 Dec;16(12):1079-85. doi: 10.1089/jmf.2012.0303. PMID: 24328700.
The authors might want to mention dairy free alternatives as well like almond milk and coconut milk. The point was made that fermented milk is usually ok. Fermented products contain the enzymes and bacteria that help with digestion.
Under the yeast section 3.1.16, sourdough bread is noted as being better tolerated than yeast based breads. A long fermentation process for sourdough bread helps with its digestibility and supporting gut microbiota. Modern wheat (hybridised) is less well tolerated than ancient grains like kamut (Khorasan wheat) with lower inflammatory response especially to gluten. Under the yeast section or under wheat in the cereals section 3.1.8, the authors could add more to support why is wheat is problematic. Ancient and heritage wheat varieties have different anti-inflammatory and antioxidant proprieties with respect to modern cultivars (Spisni et al 2019), a review recently published in Nutrients.
Spisni E, Imbesi V, Giovanardi E, Petrocelli G, Alvisi P, Valerii MC. Differential Physiological Responses Elicited by Ancient and Heritage Wheat Cultivars Compared to Modern Ones. Nutrients. 2019 Dec;11(12):2879.
Spisni has evidence on the kamut e.g. ‘During the intervention period with Khorasan wheat-based products, IBS patients showed a significant reduction in symptom severity associated with a decrease in serum levels of pro-inflammatory cytokines, such as IL-6, IL-17, interferon-γ, and MCP-1, cytokines and chemokines that in some studies were found to be higher in the IBS patient population than in healthy controls [83]. …modern grains, when clinically tested, clearly showed pro-inflammatory and pro-oxidant activities…
‘Maize’ is also known as corn and could all be summarised in the ‘Corn’ section.
Under the dairy section 3.1.5, the authors cited that ‘Calcium is necessary to prevent metabolic disease in IBD patients [51]’ This should read…prevent metabolic bone disease…
I personally think bone broth should find a spot in this review too (where the other fluids are listed?) and be part of the GrAID as it is important in the gut healing process. Bone broth is a food with ancient origins and rich in collagen, the most abundant protein in the human body. The study by Ramadass et al 2016 showed that ‘collagen and collagen hydrolysate (which is broken down collagen) significantly reduced rectal bleeding and down-regulated inflammatory markers in mice. In fact, the collagen treatments worked better than a common IBD medication, mesalamine.’
Ramadass SK, Jabaris SL, Perumal RK, HairulIslam VI, Gopinath A, Madhan B. Type I collagen and its daughter peptides for targeting mucosal healing in ulcerative colitis: A new treatment strategy. Eur J Pharm Sci. 2016 Aug 25;91:216-24. doi: 10.1016/j.ejps.2016.05.015. Epub 2016 May 13. PMID: 27185300.
